# Dependence in instrumental activities of daily living and its implications for older adults' oral health

Maria Vieira de Lima Saintrain[1]*, Suzanne Vieira Saintrain[1], July Grassiely de Oliveira Branco[1], José Manuel Peixoto Caldas[2], Caroline Barbosa Lourenço[1], Anya Pimentel Gomes Fernandes Vieira-Meyer[3,4]

**1** Public Health Graduate Program, University of Fortaleza (UNIFOR), Fortaleza, Ceará, Brazil, **2** Center for Health Technology and Services Research (CINTESIS), University of Porto (UP), Porto, Portugal, **3** Family Health Master's Program, Oswaldo Cruz Foundation, Eusébio, Ceará, Brazil, **4** Faculty of Dentistry, Centro Universitário Christus, Fortaleza, Ceará, Brazil

* mvlsaintrain@yahoo.com.br

**Data Availability Statement:** All relevant data are within the paper.

## Abstract

We aimed to assess the association between dependence in instrumental activities of daily living (IADL) and oral health in older adults. We conducted a cross-sectional study of 280 people aged ≥60 years served at public primary health care centers in Northeastern Brazil. Sociodemographic, oral discomfort and general health data were collected. The Lawton and Brody scale were used to assess IADL. This research adheres to the STROBE checklist. Most participants were married (n = 139; 49.6%), women (n = 182; 65.0%) and retired (n = 212; 75.7%). A total of 37 (13.2%) older adults had some degree of dependence in IADL. Dependence in IADL was associated with: retirement (p<0.040), poor general health (p = 0.002), speech problems (p = 0.014), use of medications (p = 0.021), difficulty chewing and swallowing food (p = 0.011), voice changes (p = 0.044), edentulism (p = 0.011), use of tooth-brush (p<0.001), use of toothpaste (p<0.001), and visit to the dentist in the previous year (p = 0.020). Functional disability was associated with older age, cardiovascular diseases, speech problems, chewing and swallowing difficulties, use of medication and brushing deficiency. The functional dependence in IADL can be considered an indicator of oral health status in older adults.

## Introduction

The decline in functional capacity during aging leads to difficulties in performing basic activities of daily living–ADL (bathing, dressing, feeding, transferring/walking and sphincter control) and instrumental activities of daily living–IADL (keeping track of finances, using the telephone or computer, managing medication, shopping) [1], where IADL involve more complex levels of physical functioning and neuropsychological organization than ADL [2].

Disability has a huge impact on oral health care and the treatment of functionally dependent older adults is challenging due to their inability to perform oral hygiene and their need to

**Funding:** The authors received no specific funding for this work.

**Competing interests:** The authors have declared that no competing interests exist.

use multiple drugs, which create an oral environment conducive to a rapid destruction of teeth [3, 4]. On the other hand, oral diseases represent a silent epidemic that disproportionately affects older people, with caries and periodontal diseases standing out as the two greatest health threats and the most common chronic diseases for the cause of tooth loss among adults [5], which in turn, has the potential to affect elders dependency, specially related with proper nutrition. For the WHO, a minimum of 20 teeth are necessary for proper oral function/nutrition [6].

Poor oral health in older people is mainly manifested in high rates of tooth loss, dental caries, periodontal diseases, xerostomia (dry mouth) and other conditions resulting from poor oral hygiene [7]. Oral diseases are undoubtedly a global public health problem linked to social and economic changes, and their ongoing neglect in global health policy highlights the need to address oral diseases among other non-communicable diseases (NCDs) as a global health priority [8].

According to the Global Burden of Disease Study 2016, oral diseases affected circa 3.58 billion people worldwide, and caries of the permanent teeth were the most prevalent of the conditions assessed. The cumulative burden of oral conditions dramatically increased between 1990 and 2015. The number of people with untreated oral conditions rose from 2.5 billion in 1990 to 3.5 billion in 2015, with a 64% increase in DALYs (disability-adjusted life year) due to oral conditions throughout the world [9]. Severe periodontal disease, which results mainly from poor oral hygiene and which can lead to tooth loss, was the 11th most prevalent disease globally [10]. Researchers have found that, in adjusted Poisson regression analysis, older age and functional disability were associated with edentulism [11]. Thus, it seems that oral health effects and is affected by elders' functionality.

The increase in life expectancy has attracted growing interest in the effect of oral health on general health outcomes, especially those related to functionality, well-being and mortality [12]. Oral health and general health share common risk factors related to diet, the use of tobacco, and the excessive consumption of alcohol and the solutions to control oral disease are to be found through shared approaches with integrated chronic disease prevention [13].

In this regard, public health solutions for oral diseases are most efficient when they are integrated with those for other NCDs and other national public health programs. The importance of oral health goals was first emphasized in 1981 by WHO as part of the program Health for All by the year 2000 [14, 15] and most recently the WHO, jointly with the FDI World Dental Federation and the International Association for Dental Research (IADR), formulated goals for oral health by the year 2020 [16].

Furthermore, the WHO Global Oral Health Program is aligned with the Global NCD agenda and the Shanghai Declaration on promoting health in the 2030 Agenda for Sustainable Development [17]. According to Petersen [13], the WHO Oral Health program gives priority to research helping correct the so called 10/90 gap which relates to the fact that only 10%of funding for global health research is allocated to health problems that affect 90% of the world population. The WHO Oral Health Program focuses on stimulating oral health research in the developed and developing world to reduce risk factors and the burden of oral disease, and to improve oral health systems and the effectiveness of community oral health programs. Building and strengthening research capacity in public healthcare highly recommended by WHO for effective control of disease and the socioeconomic development of any given country.

In Brazil, the rate of dependence in at least one ADL or IADL among older adults is 32.6% and the prevalence rate of exclusive dependence in IADL is 14.6% [18]. Population aging can be categorized into three broad functional groups: functionally independent, frail, and functionally dependent. Chronic and oral health diseases have a greater potential to impact frail

and functionally dependent older people. However, it is important to understand that, although chronic illness is common among older adults, it does not necessarily lead to disability, but constitutes an important risk factor for its development [19].

In view of what was outlined above, there is an evident need for collaboration between the medical and dental fields and between professionals in both fields in order to make healthcare provision systems more effective [20]. Therefore, understanding the dimensions of the impact in terms of functional impairment caused by poor oral and general health is critical for the provision of adequate health care, communication and health education [21]. Given that, the objective of the present study was to assess the relationship between IADL and Oral Health of older people.

## Materials and methods

All procedures performed in studies involving human participants were in accordance with the national research committee, as per approval No. 200/2009 from the University of Fortaleza Institutional Review Board. This committee follows the 1964 Helsinki declaration and its later amendments or comparable ethical standards. Informed consent was obtained from all individual participants included in the study.

The present analytical and descriptive cross-sectional study is part of a larger study titled "physical and mental performance tests as tools for the assessment and identification of oral health in older patients" funded by Brazil's National Council for Scientific and Technological Development (*Conselho Nacional de Desenvolvimento Científico e Tecnológico–CNPq*) under process number: 478645/2013–9.

The research sample consisted of older people (60 years old and older) assisted by the Primary Health Care System of the city of Fortaleza. The city of Fortaleza is the capital of the state of Ceará in Northeastern Brazil. It is the fifth largest city in the country with approximately 2.6 million inhabitants and is divided into six administrative regions which are called Regional Executive Secretariats (*Secretaria Executiva Regional–SER*) [22].

All the participants were informed of the objectives and procedures of the study. Participants anonymity and confidentiality of information were guaranteed. Data were individually collected and analyzed using:

1. Semi-structured questionnaire addressing sociodemographic data (age, sex, race, marital status, education and income), general health (poor, fair, very good), self-reported diseases and oral discomfort data (dry mouth, difficulty in chewing and swallowing food, taste of food, burning mouth sensation, pain without apparent reason, swelling of the mouth), as well as alcohol and smoking habits.

2. Community Oral Health Indicator (*Indicador Comunitário de Saúde Bucal–ICSB)*. This validated indicator checks the number of teeth, visible dental cavities, presence of calculus, gingival inflammation, residual root, soft tissues injuries, use and need of dental prosthesis, use of toothbrush and toothpaste, and visits to the dentist [23]. The examinations were performed under natural light with the help of a wooden spatula (tongue depressor)–examiners wore personal protective equipment.

3. Instrumental activities of daily living (IADL)–IADL were assessed by the Lawton and Brody Scale [24]. This instrument includes eight variables related to mobility (using the telephone, getting to distant places using some mode of transportation, shopping, housekeeping, doing the laundry, preparing own food, taking medications and handling finances). Each item is rated according to three alternatives: Independent (3 points), Needs assistance (2 points) and Unable (1 point). The maximum score is 27. Individuals with

scores of 19–27 points are classified as independent. Scores of 10–18 indicate assisted independence and scores of 1–9 points indicate dependence.

The sample size was estimated considering the number of older adults (N = 242,430) in the city of Fortaleza in the year 2012 as reported in Brazil's National Health System Database [25]. A minimum sample size was estimated considering an expected maximum proportion of 20%, a significance level of 5% (95% confidence interval), and a maximum permissible error of 5%. The following formula for finite populations was used:

$$n = \frac{N \cdot k^2 \cdot p \cdot q}{e^2 \cdot (N-1) + k^2 \cdot p \cdot q}$$

Where: n = sample size; N = population; k = parameter of significance level; p = % of occurrence; q = difference of 1 –p; e = sampling error.

The formula estimated a minimum sample size of 246 older people.

The Statistical Package for the Social Sciences–SPSS version 21 (SPSS Inc., Chicago, IL, USA) was used for data analysis. Pearson's Chi-squared test[1], Fisher's Exact test[2], and Prevalence Ratio (PR) were used with a significance level set at 5% to test the hypothesis.

## Results

The study sample consisted of 280 older adults ranging from 60–96 years of age, with a mean age of 69.9 (SD±7.2) years. Table 1 shows that most participants were 60–69 years old (n = 154; 55.0%), married (n = 139; 49.6%), women (n = 182; 65.0%), *pardos* (mixed-race Brazilians whose color is intermediate between "black" and "white") (n = 134; 47.9%), retired (n = 212; 75.7%), had a monthly income of up to one minimum wage and had completed primary education. A total of 37 (13.2%) participants had some degree of dependence in IADL. Dependence in IADL differed significantly between age groups (p<0.001)[2] and was associated with being retired (p = 0.040)[2] and education (p = 0.042)[2].

Table 2 shows that dependence in IADL was associated with general health status (p = 0.002)[2], cardiovascular disorders (p = 0.039)[1], speech problems (p = 0.014)[1], use of medications (p = 0.021)[2] and drinking (p = 0.038)[2]. Prevalence ratio estimates showed that dependent older people were 4.13 times more likely to present poor general health, 1.90 times more likely to have cardiovascular disorders, 2.51 times more likely to have speech problems and 2.61 more likely to use medications. Non-smokers and non-drinkers presented 2.06 and 5.65, respectively, more chances of not being dependent.

Table 3 shows that dependence in IADL was also significantly associated with chewing and swallowing difficulties (p = 0.011)[2], voice changes (p = 0.044)[2], edentulism (p = 0.011)[2], number of teeth (p = 0.021)[2], use of toothbrush (p<0.001)[2], use of toothpaste (p<0.001)[2], and visit to the dentist in the previous year (p = 0.020)[2]. Dependent older people were 2.19 times more likely to present chewing and swallowing difficulties and 2.22 times more likely to have voice changes. Older people who used toothpaste had 5.07 times more chances of not being dependent and those who did not have dental visits in the previous year were 2.42 times more likely to be dependent.

Table 4 shows that very old adults presented 8.66 times more chances of being dependent in IADL than older adults aged 60–69 years in the Poisson regression model. In addition, older adults with cardiovascular disorders were 1.77 more likely to be dependent in IADL than those without cardiovascular disorders and older adults who did not use toothbrush were 2.57 times more likely to be dependent in IADL than those who used toothbrush.

**Table 1. Frequency distribution and inferential analysis of older people's sociodemographic data versus dependence and independence in instrumental activities of daily living (IADL).**

| Variables | Dependence | Independence | PR (95%CI) | p value |
|---|---|---|---|---|
| | n (%) | | | |
| **Age group** | | | | |
| 80 or older | 16 (43.2) | 17 (7.0) | 12.44 (5.27–29.4) | <0.001[2] |
| 70–79 years | 15 (40.5) | 78 (32.1) | 4.14 (1.66–10.30) | |
| 60–69 years | 6 (16.2) | 148 (60.9) | 1 | |
| **Marital Status** | | | | |
| Married | 17 (45.9) | 124 (51.0) | 5.06 (0.69–36.94) | 0.027[2] |
| Divorced | 4 (10.8) | 20 (8.2) | 7.00 (0.83–59.09) | |
| Widowed | 15 (40.5) | 58 (23.9) | 8.63 (1.18–63.03) | |
| Single | 1 (2.7) | 41 (16.9) | 1 | |
| **Sex** | | | | |
| Male | 13 (35.1) | 85 (35.0) | 1.01 (0.54–1.89) | 1.000[1] |
| Female | 24 (64.9) | 158 (65.0) | 1 | |
| **Race** | | | | |
| White | 15 (40.5) | 108 (44.4) | 1.22 (0.30–4.93) | 0.789[2] |
| *Pardo* | 20 (54.1) | 117 (48.1) | 1.46 (0.37–5.78) | |
| Black | 2 (5.4) | 18 (7.4) | 1 | |
| **Retired** | | | | |
| Yes | 33 (89.2) | 179 (73.7) | 2.65 (0.97–7.20) | 0.040[2] |
| No | 4 (10.8) | 64 (26.3) | 1 | |
| **Income** | | | | |
| Less than one MW | 4 (10.8) | 33 (13.6) | 1 | 0.898[2] |
| 1 MW | 26 (70.3) | 157 (64.6) | 1.31 (0.49–3.54) | |
| 2–5 MW | 6 (16.2) | 49 (20.2) | 1.01 (0.31–3.33) | |
| More than 5 MW | 1 (2.7) | 4 (1.6) | 1.85 (0.25–13.43) | |
| **Education** | | | | |
| None | 11 (31.4) | 42 (17.3) | 2.49 (0.85–7.30) | 0.042[2] |
| Incomplete primary | 18 (51.4) | 103 (42.4) | 1.79 (0.64–5.00) | |
| Inomplete secondary | 4 (8.6) | 54 (22.2) | 0.83 (0.22–3.14) | |
| Complete secondary / Higher education | 4 (8.6) | 44 (18.1) | 1 | |

Source: research data. MW = minimum wage (Brazilian minimum wage is R$ 1,100.00—approximately U$ 200.00).

[1]Pearson's Chi-squared test

[2]Fisher's Exact test, Odds Ratio (OR), Confidence Interval = 95%CI

## Discussion

The present study stands out for showing a relationship between dependence in IADL and oral health of older adults. Gender did not appear to have any influence on IADL dependency. This finding differs from a study that demonstrated that women have a greater incidence of disability in IADL than men, and that this difference is maintained even after controlling for social vulnerability among women and presence of chronic diseases [2]. According to the latest IBGE census, women represent 60,7% of the elder population in Fortaleza [26]. Thus, through authors" perspective, our female sample (65%) it comparable with the city reality and did not seem to influence our findings regarding IADL in the elder population.

In the present study, the most prevalent age group and the mean age are similar to those of another Brazilian study, where the mean age was 70.04 (±7.89) years and most people were

**Table 2. Frequency distribution and inferential analysis of older people's general health versus dependence and independence in instrumental activities of daily living (IADL).**

| Variables | Dependence | Independence | PR (95%CI) | p value |
|---|---|---|---|---|
| | n (%) | | | |
| **General Health** | | | | 0.002[2] |
| Poor | 14 (37.8) | 36 (14.8) | 4.13 (1.45–11.75) | |
| Fair | 19 (51.4) | 152 (62.6) | 1.64 (0.58–4.62) | |
| Very good | 4 (10.8) | 55 (22.6) | 1 | |
| **Diabetes mellitus** | | | | 0.146[1] |
| Yes | 14 (37.8) | 64 (26.3) | 1.58 (0.86–2.90) | |
| No | 23 (62.2) | 179 (73.7) | 1 | |
| **Cardiovascular disorders** | | | | 0.039[1] |
| Yes | 23 (62.2) | 107 (44.0) | 1.90 (1.02–3.53) | |
| No | 14 (37.8) | 136 (56.0) | 1 | |
| **Osteoporosis** | | | | 0.340[1] |
| Yes | 10 (27.0) | 49 (20.2) | 1.39 (0.71–2.70) | |
| No | 27 (73.0) | 194 (79.8) | 1 | |
| **Rheumatism** | | | | 0.596[2] |
| Yes | 6 (16.2) | 30 (12.3) | 1.31 (0.59–2.92) | |
| No | 31 (83.8) | 213 (87.7) | 1 | |
| **Vision problems** | | | | 0.373[1] |
| Yes | 27 (73.0) | 193 (79.4) | 1 | |
| No | 10 (27.0) | 50 (20.6) | 1.36 (0.70–2.65) | |
| **Hearing problems** | | | | 0.220[1] |
| Yes | 14 (37.8) | 68 (28.0) | 1.47 (0.80–2.71) | |
| No | 23 (62.2) | 175 (72.0) | 1 | |
| **Speech problems** | | | | 0.014[1] |
| Yes | 10 (27.0) | 26 (10.7) | 2.51 (1.33–4.74) | |
| No | 27 (73.0) | 217 (89.3) | 1 | |
| **Use of medications** | | | | 0.021[2] |
| Yes | 32 (86.5) | 167 (68.7) | 2.61 (1.05–6.45) | |
| No | 5 (13.5) | 76 (31.3) | 1 | |
| **Smoking** | | | | 0.189[2] |
| Yes | 3 (8.1) | 40 (16.5) | 1 | |
| No | 34 (91.9) | 203 (83.5) | 2.06 (0.66–6.40) | |
| **Drinking** | | | | 0.038[2] |
| Yes | 1 (2.7) | 37 (15.2) | 1 | |
| No | 36 (97.3) | 206 (84.8) | 5.65 (0.80–40.03) | |

Source: research data.

[1]Pearson's Chi-squared test

[2]Fisher's Exact test, Odds Ratio (OR), Confidence Interval = 95%CI

aged 60–69 years [27]. The percentage of 13.2% of older people with different degrees of dependence in their IADL is similar to that reported in the literature for Brazil [18, 28]. However, a study shows that 46.3% of a population of 289 older people had moderate to severe dependence in IADL [29]. It is important to know that the characteristics identified regarding dependence in IADL suggest a complex causal relationship that points to the need for preventive actions specifically designed to address different factors and improve older people's quality of life [27]. Nonetheless, it is important to point out that all the studies just mentioned were

**Table 3. Frequency distribution and inferential analysis of older people's oral health status and oral discomfort variables versus dependence and independence in instrumental activities of daily living (IADL).**

| Variables | Instrumental Activities of Daily Living | | PR (95%CI) | *p* value |
|---|---|---|---|---|
| | Dependence | Independence | | |
| | n (%) | n (%) | | |
| **Number of teeth** | | | | 0.021[1] |
| None | 26 (70.3) | 116 (47.7) | 1.98 (1.02–3.84) | |
| 1–19 teeth | 11 (29.7) | 108 (44.4) | 1 | |
| 20 or more | - | 19 (7.8) | - | |
| **Edentulous** | | | | 0.011[1] |
| Yes | 26 (70.3) | 116 (47.7) | 2.30 (1.18–4.47) | |
| No | 11 (29.7) | 127 (52.3) | 1 | |
| **Presence of calculus** | | | | 0.201[1] |
| Yes | 9 (24.3) | 85 (35) | 1 | |
| No | 28 (75.7) | 158 (65) | 1.57 (0.77–3.19) | |
| **Gingival inflammation** | | | | 0.172[2] |
| Yes | 3 (8.1) | 41 (16.9) | 1 | |
| No | 34 (91.9) | 202 (83.1) | 2.11 (0.68–6.58) | |
| **One or two visible cavities** | | | | 0.102[2] |
| Yes | 2 (5.4) | 26 (10.7) | 1 | |
| No | 35 (94.6) | 217 (89.3) | 1.94 (0.49–7.66) | |
| **Three or more visible cavities** | | | | 1.000[2] |
| Yes | 1 (2.7) | 6 (2.5) | 1.08 (0.17–6.82) | |
| No | 36 (97.3) | 237 (97.5) | 1 | |
| **Soft tissue injuries** | | | | 0.441[1] |
| Yes | 11 (29.7) | 58 (23.9) | 1.29 (0.68–2.48) | |
| No | 26 (70.3) | 185 (76.1) | 1 | |
| **Need for prosthesis** | | | | 0.732[1] |
| Yes | 20 (54.1) | 124 (51.0) | 1.11 (0.61–2.03) | |
| No | 17 (45.9) | 119 (49.0) | 1 | |
| **Use of prosthesis** | | | | 0.070[1] |
| Yes | 23 (62.2) | 185 (76.1) | 1 | |
| No | 14 (37.8) | 58 (23.9) | 1.76 (0.96–3.23) | |
| **Use of toothbrush** | | | | <0.001[1] |
| Yes | 25 (67.6) | 230 (94.7) | 1 | |
| No | 12 (32.4) | 13 (5.3) | 4.90 (2.82–8.51) | |
| **Use of toothpaste** | | | | <0.001[1] |
| Yes | 23 (62.2) | 227 (93.4) | 1 | |
| No | 14 (37.8) | 16 (6.6) | 5.07 (2.94–8.76) | |
| **Dental visit in the past year** | | | | 0.020[1] |
| Yes | 7 (18.9) | 94 (38.7) | 1 | |
| No | 30 (81.1) | 149 (61.3) | 2.42 (1.10–5.31) | |
| **Difficulty chewing and swallowing** | | | | 0.011[1] |
| Yes | 14 (37.8) | 47 (19.3) | 2.19 (1.20–3.98) | |
| No | 23 (62.2) | 196 (80.7) | 1 | |
| **Problems with taste** | | | | 0.164[1] |
| Yes | 9 (24.3) | 37 (15.2) | 1.64 (0.83–3.23) | |
| No | 28 (75.7) | 206 (84.8) | 1 | |
| **Pain with no apparent reason** | | | | 0.614[1] |

(*Continued*)

**Table 3.** (Continued)

| Variables | Instrumental Activities of Daily Living | | PR (95%CI) | *p* value |
|---|---|---|---|---|
| | Dependence | Independence | | |
| | n (%) | n (%) | | |
| Yes | 6 (16.2) | 32 (13.2) | 1.23 (0.55–2.76) | |
| No | 31 (83.8) | 211 (86.8) | 1 | |
| **Voice changes** | | | | 0.044[1] |
| Yes | 8 (21.6) | 23 (9.5) | 2.22 (1.11–4.41) | |
| No | 29 (78.4) | 220 (90.5) | 1 | |

Source: research data.

[1]Pearson's Chi-squared test

[2]Fisher's Exact test, Odds Ratio (OR), Confidence Interval = 95%CI

carried out in Brazil. As the structures, clinical supports and family care of older adult populations may vary in terms of national context and culture, it is important to have these issues in mind when extrapolating the study findings.

The findings of the present research, which show that dependent older people are 4.13 times more likely to rate their general health as poor, 1.90 times more likely to have cardiovascular disorders, 2.51 times more likely to have speech problems and 2.61 times more likely to use medication, point to the need to reduce the level of exposure to important risk factors given that oral disease prevention needs to be integrated with that of chronic diseases on the basis of common risk factors [30].

Furthermore, the increased PR estimates between dependence in IADL and difficulty chewing and swallowing (2.19), voice changes (2.22), edentulous (2.30), use of toothbrush (4.90), use of toothpaste (5.07) and visit to the dentist in the previous year (2.42) demonstrates a close relationship between disability and oral health preventive measures. Our voice is an extremely sensitive indicator of our general health and emotional status [31]. The voice box (larynx) is made of cartilage, muscle and mucous membranes located at the top of the trachea and the base of the tongue. If the vocal cords become inflamed, develop growths or become paralyzed, they can't work properly, and one may develop a voice disorder. A voice change can be a sign of throat dehydration, Neurological disorders, or consequence of habits like alcohol consumption and tobacco usage. Furthermore, voice change can also be linked with oral discomfort. Considering that dental caries and periodontal disease are associated with lifestyle, patients

**Table 4. Older people's variables that remained associated with dependence in instrumental activities of daily living (IADL) in the Poisson regression model.**

| Variables | Adjusted PR | 95%CI | *p* value |
|---|---|---|---|
| **Age group** | | | |
| 80 or older | 8.66 | 3.48–21.57 | <0.001 |
| 70–79 years | 3.71 | 1.48–9.30 | 0.005 |
| 60–69 years | 1 | | |
| **Cardiovascular disorders** | | | |
| Yes | 1.77 | 1.01–3.08 | 0.045 |
| No | 1 | | |
| **Use of toothbrush** | | | |
| Yes | 1 | | |
| No | 2.57 | 1.45–4.56 | 0.001 |

should be encouraged to visit the dentist regularly in addition to being provided with dental health guidance in each visit, as health education has short-term effects [32].

The fact that age, cardiovascular disorders and use of toothbrush remained associated with dependence in IADL in the Poisson regression model is in line with the findings of other studies. Researchers found a significant association between dependence in IADL and age $\geq 80$ years in a multiple multinomial Poisson model (p = 0.007), which may be a result of the greater decline in the nervous and musculoskeletal systems in very old adults [33]. Also, there is a significant association between cardiovascular disorders and dependence in IADL, which may be explained by the potential disability and sequelae resulting from heart disease [34]. Cardiovascular disorders may lead to motor impairments, such as loss of coordination, which, although not assessed in our study, may affect the handling and use of toothbrush and, therefore, impact oral health [35].

In the past century, researchers emphasized that oral diseases were rarely seen in terms of their impact on patient's functionality and that even oral health problems were not considered in the geriatric assessment [36]. Oral health was traditionally assessed without considering the general health status and it was often considered to be unrelated to other chronic diseases or conditions. However, the relationship between functional dependence and maintenance of oral health seems to be evident [37].

We assessed the impact of functionality on oral health in a previous study, in which we analyzed the relationship between dependence in ADL and oral health status in older adults. It was revealed that functional dependence in ADL has implications for the oral health status of older people [38]. Therefore, assessing the relationship between dependence in IADL and oral health status can deepen the knowledge of the impact of older people's functionality on oral health and contribute to the development of public policies and individual care plans for this population through interdisciplinary and interprofessional collaboration.

In this context, difficulties in performing motor tasks, such as hand manipulation, caused by the decrease in grip strength may affect several IADL [1, 21]. Consequently, difficulties in performing activities such as tooth brushing and body hygiene may directly affect the oral and general health maintenance.

According to the World Health Organization, oral diseases have a significant adverse impact on general health and quality of life, and a healthy and well-functioning dentition is important during all stages of life to support essential human functions, such as speaking, smiling, socializing and eating [7]. In addition, for frail and functionally dependent (community-dwelling, institutionalized and homebound) older people, the involvement of trained health care providers and caregivers plays a key role in oral and general health programs regarding diet and nutrition. Therefore, in order to promote healthy aging, health systems should share a multisectoral approach that promotes the development of a health-friendly environment [21].

Based on the current evidence, it is important to consider that the contributions to general health, made by oral health, should be clarified in order to highlight the effects of oral health on various factors affecting general health, which in turn, demands the development of new policy proposals in the area [39].

The findings of the present research showed that functional dependence in IADL can have a direct impact on oral hygiene and behaviors that maintain oral health, and therefore, is an important health indicator in older adults. In this regard, researchers have demonstrated that an educational program tailored to caregivers had a positive impact on the oral health of dependent older adults, observed by the improved oral hygiene parameters, such as plaque index and tongue coating [40].

Oral health problems can in turn have profound effects on general health and well-being. Furthermore, oral pain and problems with eating, chewing, smiling and communicating due

to missing, discolored or damaged teeth have a major impact on functional ability and older people's daily lives [21].

It should be noted that the present study was carried out in one single city. Therefore, its results cannot be extrapolated, which represents a limitation of the study. However, the study was conducted in a large city in Brazil and its results are likely to be found in other places. Despite that, complementary and more in-depth studies should be conducted to clarify the relationship of functional dependence and its determining factors with oral health status.

## Conclusion

Our findings demonstrated that functional dependence in IADL is related to oral health status in older Brazilian adults. Functional disability was associated with older age, cardiovascular diseases, speech problems, chewing and swallowing difficulties, use of medication and brushing deficiency. Therefore, health promotion and disease prevention and rehabilitation actions should be carried out at all levels of health care considering the needs of this population group in order to improve oral hygiene and oral health-friendly behaviors.

## Author Contributions

**Conceptualization:** Maria Vieira de Lima Saintrain, Anya Pimentel Gomes Fernandes Vieira-Meyer.

**Data curation:** Maria Vieira de Lima Saintrain, July Grassiely de Oliveira Branco, Caroline Barbosa Lourenço.

**Formal analysis:** Maria Vieira de Lima Saintrain, Suzanne Vieira Saintrain, José Manuel Peixoto Caldas, Anya Pimentel Gomes Fernandes Vieira-Meyer.

**Methodology:** Maria Vieira de Lima Saintrain, José Manuel Peixoto Caldas.

**Project administration:** Maria Vieira de Lima Saintrain, Anya Pimentel Gomes Fernandes Vieira-Meyer.

**Supervision:** Maria Vieira de Lima Saintrain, José Manuel Peixoto Caldas, Anya Pimentel Gomes Fernandes Vieira-Meyer.

**Visualization:** July Grassiely de Oliveira Branco, Caroline Barbosa Lourenço.

**Writing – original draft:** Maria Vieira de Lima Saintrain, Suzanne Vieira Saintrain, July Grassiely de Oliveira Branco, José Manuel Peixoto Caldas, Caroline Barbosa Lourenço.

**Writing – review & editing:** Maria Vieira de Lima Saintrain, Suzanne Vieira Saintrain, July Grassiely de Oliveira Branco, José Manuel Peixoto Caldas, Caroline Barbosa Lourenço.

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
