## [Decision Letter · Decision Letter 0]

21 Sep 2020

PONE-D-20-18747

Dependence in instrumental activities of daily living and its implications for older adults’ oral health

PLOS ONE

Dear Dr. Saintrain,

Thank you for submitting your manuscript to PLOS ONE. After careful consideration, we feel that it has merit but does not fully meet PLOS ONE’s publication criteria as it currently stands. Therefore, we invite you to submit a revised version of the manuscript that addresses the points raised during the review process.

We look forward to receiving your revised manuscript.

Kind regards,

Frédéric Denis, Ph.D.

Academic Editor

PLOS ONE

Journal Requirements:

2. Please provide additional details regarding participant consent. In the Methods section, please ensure that you have specified what type of consent you obtained (for instance, written or verbal) and whether the ethics committee approved this consent procedure. If verbal consent was obtained please state why it was not possible to obtain written consent and how verbal consent was recorded. If your study included minors, state whether you obtained consent from parents or guardians.

3. Thank you for including your ethics statement:  "All procedures performed in studies involving human participants were in accordance with the Institutional Review Board (Approval No. 200/2009) and the national research committee and with the 1964 Helsinki declaration and its later amendments or comparable ethical standards. Informed consent was obtained from all individual participants included in the study.".   

Please amend your current ethics statement to confirm that your named institutional review board or ethics committee specifically approved this study.

Reviewers' comments:

Reviewer's Responses to Questions

**Comments to the Author**

1. Is the manuscript technically sound, and do the data support the conclusions?

Reviewer #1: Partly

Reviewer #2: Yes

2. Has the statistical analysis been performed appropriately and rigorously? 

Reviewer #1: I Don't Know

Reviewer #2: I Don't Know

3. Have the authors made all data underlying the findings in their manuscript fully available?

Reviewer #1: Yes

Reviewer #2: No

4. Is the manuscript presented in an intelligible fashion and written in standard English?

Reviewer #1: Yes

Reviewer #2: Yes

5. Review Comments to the Author

Reviewer #1: Oral health is an understudied aspect of health, and additions to the scientific literature are needed. However, I identified a number of weaknesses in the manuscript that, without being addressed, substantively limit the degree to which the manuscript adds to the knowledge base.

p. 3, paragraph 2: The first 2 sentences don’t clearly connect to previous statements or the sense of the paragraph – they seem to be addressing different sentiments that need separate paragraphs… There are other problems with that paragraph, in terms of apparent disjointed ideas appearing. For example, the statement about associations with edentulism being followed by a statement about the prevalence rate of dependence in IADL, with no transitional connection between them.

As I read along further, a more general issue became apparent. Specifically, there is a lack of theoretical framework presented to tie together oral health and IADL. What is the mechanism that ties those factors together? The theoretical framework providing the impetus for this study must be made explicit to support the study premise. This is especially needed due to the causal flow implied by your choice of outcome and independent variables. For example, on p. 10, you indicate that “…older people who used toothpaste had 5.07 times more chances of not being dependent …”. Are you really saying that the causal flow is toothpaste use � dependence? It seems to me to be at least as likely (actually, more likely) that independence � toothpaste use.

Specific hypotheses or, at the very least, research questions, are not presented, which, combined with the scant statistical analysis description presented, makes evaluation of statistical method appropriateness difficult, especially with regards to Table 4.

Sample size justification was apparently predicated on contingency table analyses planned – justification is needed that the parameters used are realistic and represent clinically substantive associations.

Justification is needed for the cut-points used to produce ordered categories – for example, citation for the cut-points defining independence groups; rationale for the age groupings utilized, etc.

Is there information related to the validity and reliability of the Lawton and Brody scale in a Brazilian population? How about validity and reliability of selected other measures (e.g.: general health items, oral health items)?

I think the discussion of recommendations with regards to oral health for patients who are dependent (p. 13) may be rather simplistic. Given that such patients are dependent on others (caregivers), it seems like one explanation would be that the dependence/disability is the cause of the poor oral hygiene and oral health, and that the caregivers, might need intervention as much as the patient… That reasoning does come out on page 14.

I was confused about the discussion at the bottom of page 15, where you talk about a program tailored for caregivers – are you saying that the positive findings for that study support an underlying premise that poor oral health in dependent patients may be due to inadequate oral hygiene provided by caregivers? That is a logical conclusion, given that the dependent patient is dependent for care on others – just need to explain what you mean more clearly…

Reviewer #2: Thank you for your submission and yes I agree that we need to assess the status of older population in order to better serve their needs.

You mentioned that some of your questionnaires are validated what about the others, where they validated? If not, how can you justify their use and reliability?

-Were your examiners calibrated? If yes, how and how is their agreement. If not, how reliable is your data?

-Where you able to match data from supposes? How does marriage play a role in assessing dependance?

-There were some instances that you used causation as a term instead of association. Causation could not be established at this stage so please correct accordingly.

-While you addressed generalizability as a limitation but it was not part of your conclusion statement. Please modify your conclusion so that it is clear that it does not apply to all population.

6. PLOS authors have the option to publish the peer review history of their article (what does this mean?). If published, this will include your full peer review and any attached files.

Reviewer #1: No

Reviewer #2: No

---

## [Author Response · Author response to Decision Letter 0]

2 Nov 2020

Below the answers for reviewers questions - point-by-point coments can be found right after each reviewer comment. 

Reviewer's Responses to Questions

Comments to the Author

1. Is the manuscript technically sound, and do the data support the conclusions?

Reviewer #1: Partly

Reviewer #2: Yes

2. Has the statistical analysis been performed appropriately and rigorously? 

Reviewer #1: I Don't Know

Reviewer #2: I Don't Know

3. Have the authors made all data underlying the findings in their manuscript fully available?

Reviewer #1: Yes

Reviewer #2: No

4. Is the manuscript presented in an intelligible fashion and written in standard English?

Reviewer #1: Yes

Reviewer #2: Yes

5. Review Comments to the Author

Reviewer #1: Oral health is an understudied aspect of health, and additions to the scientific literature are needed. However, I identified a number of weaknesses in the manuscript that, without being addressed, substantively limit the degree to which the manuscript adds to the knowledge base.

We appreciate your comments and suggestions. Adjustments were performed in the manuscript in order to address the issues raised by your revisions. 

p. 3, paragraph 2: The first 2 sentences don’t clearly connect to previous statements or the sense of the paragraph – they seem to be addressing different sentiments that need separate paragraphs… There are other problems with that paragraph, in terms of apparent disjointed ideas appearing. For example, the statement about associations with edentulism being followed by a statement about the prevalence rate of dependence in IADL, with no transitional connection between them.

As I read along further, a more general issue became apparent. Specifically, there is a lack of theoretical framework presented to tie together oral health and IADL. What is the mechanism that ties those factors together? The theoretical framework providing the impetus for this study must be made explicit to support the study premise. 

A new paragraph was inserted in the text, as well as a reorganization of the previous paragraph was performed in order to address the issues raised by the reviewer. Below is a transcription of the actual text.

“The decline in functional capacity during aging leads to difficulties in performing basic activities of daily living – ADL (bathing, dressing, feeding, transferring/walking and sphincter control) and instrumental activities of daily living – IADL (keeping track of finances, using the telephone or computer, managing medication, shopping) [1]. IADL involve more complex levels of physical functioning and neuropsychological organization than ADL [2].

The literature has shown an association between chronical disease and AIDL and ADL (Nunes et al., 2010), as well as between systemic disease and oral health (Kim and Amar, 2006). Periodontal disease, for instance, has been associated with cardiovascular disease, type 2 diabetes mellitus, adverse pregnancy outcomes, such as low birth weight, and osteoporosis (Kim and Amar, 2006). Thus, oral health may be indicative of systemic health, which in turn, is related to AIDL and ADL. However, the relationship between oral health and AIDL and ADL is not commonly studied, especially in the elder population. This is a relevant issue to be evaluated, as oral health should be internalized as an important health aspect of the elder. The recognition of a bidirectional relationship between oral health and AIVD, as seen between oral health and systemic health, may improve the assistance given to elders. 

In Brazil, the rate of dependence in at least one activity of daily living (ADL) or instrumental activity of daily living (IADL) among older adults is 32.6% and the prevalence rate of exclusive dependence in IADL is 14.6% [7]. Elders can be categorized into three broad functional groups: functionally independent, frail, and functionally dependent. Although chronic illness is common among older adults, it does not necessarily lead to disability, but constitutes an important risk factor for its development [3]. Disability has a huge impact on oral health care and the treatment of functionally dependent older adults is challenging due to their inability to perform oral hygiene and their need to use multiple drugs, which create an oral environment conducive to a rapid destruction of teeth [4, 5]. Additionally, researchers have found that, in adjusted Poisson regression analysis, older age and functional disability were associated with edentulism [6].” 

This is especially needed due to the causal flow implied by your choice of outcome and independent variables. For example, on p. 10, you indicate that “…older people who used toothpaste had 5.07 times more chances of not being dependent …”. Are you really saying that the causal flow is toothpaste use � dependence? It seems to me to be at least as likely (actually, more likely) that independence � toothpaste use.

We thank the reviewer for bringing up this important point. We have adjusted the manuscript in order to clarified this issue. The objective of the present study was to evaluate the existence of an association between oral health and AIVD, which we found. At this point, we cannot imply causality in this association, and the text was adjusted in this manner. As for the text in page 10, what we are saying is that use toothpaste is associated with not being dependent. The text has been modified to the following: 

“Table 3 shows that dependence in IADL was also significantly associated with chewing and swallowing difficulties (p=0.011)2, voice changes (p=0.044)2, edentulism (p=0.011)2, number of teeth (p= 0.021)2, use of toothbrush (p<0.001)2, use of toothpaste (p<0.001)2, and visit to the dentist in the previous year (p=0.020)2. Other associations with oral issues were also identified. Dependent older people were 2.19 times more likely to present chewing and swallowing difficulties and 2.22 times more likely to have voice changes. Older people who used toothpaste had 5.07 times less chances of being dependent and those who did not have dental visits in the previous year were 2.42 times more likely to be dependent.”

Specific hypotheses or, at the very least, research questions, are not presented, which, combined with the scant statistical analysis description presented, makes evaluation of statistical method appropriateness difficult, especially with regards to Table 4.

We thank the reviewer for asking for a clear state on the research question in the manuscript. For us, it was so clear in our minds that we believed it to be clear in the article as well. As for reviewer suggestion, we have added the following text in the manuscript: “Thus, we wanted to know if oral, general health and chronicles issues were related to instrumental activities of daily living in the studied population. Furthermore, we wanted to investigate the variables that remained associated with AIVD when the associated issues were evaluated together, point out to indicators that might be use in the elder population.” 

Sample size justification was apparently predicated on contingency table analyses planned – justification is needed that the parameters used are realistic and represent clinically substantive associations.

We were not sure about the issue raised by the reviewer regarding our sample size calculation. We utilized common and broadly used parameters for the sample size calculation for finite populations, as follows: 

The sample size was estimated considering the number of older adults (N=242,430) in the city of Fortaleza in the year 2012 as reported in Brazil’s National Health System Database [18]. A minimum sample size was estimated considering an expected maximum proportion of 20%, a significance level of 5% (95% confidence interval), and a maximum permissible error of 5%. The following formula for finite populations was used: 

n = ___N . k2 . p . q_____

 e2 . (N-1) + k2 . p .q 

Where: n= sample size; N= population; k= parameter of significance level; p= % of occurrence; q= difference of 1 – p; e= sampling error. 

The formula estimated a minimum sample size of 246 older people. 

Justification is needed for the cut-points used to produce ordered categories – for example, citation for the cut-points defining independence groups; rationale for the age groupings utilized, etc.

The cut-points for AIVD on dependente, semi-dependent and independente were based on Pinto et al. (2016) classification. This has been added to the manuscript as follows: “The maximum score is 27. Individuals with scores of 19-27 points are classified as independent. Scores of 10-18 indicate assisted independence and scores of 1-9 points indicate dependence. (Pinto et al., 2016)”. 

As for age, we know that the age pyramid is commonly classified for every five years, however, in order to facilitate inferential analysis, the age of the elderly was categorized into the age groups of 60-69, 70-79, 80 and more, following the example of Sousa et al., 2018. [18]. It is important to know that in Brazil, elders older than 80years are considered an “extra” elder group and have especial civic rights. The manuscript was adjusted to insert the clarification for the age cut point as follows: “To facilitate inferential analysis, the age of the elderly was categorized into the age groups of 60-69, 70-79, 80 and more, following the example of Sousa et al., 2018. [18]”. 

Is there information related to the validity and reliability of the Lawton and Brody scale in a Brazilian population? How about validity and reliability of selected other measures (e.g.: general health items, oral health items)?

According to Santos & Virtuoso Júnior (2008), who analyzed the reliability of the Scale of Instrumental Activities of Daily Living (IADL) developed by Lawton and Brody and adapted to Brazilian context. Concerning the stability of the measures, they can be classified as almost perfect agreement, both for the reliability (Ricc=0.89) as for the objectivity (Ricc=0.80). Oral health was measured utilizing the ICSB, which was created for the Brazilian community by one of the co-authors (Saintrain, 2007) of the present manuscript and has been used in a series of studies (Saintrain & Vieira, 2012; Saintrain et al., 2018a; Saintrain et al., 2018b). Other general and oral health items, as performed in several other studies, were directly asked to the study population. Validity and reliability are not commonly measure for this type of questions.

ICSB References:

Saintrain, MVL. Proposta de um indicador comunitário em saúde bucal. RBPS 2007; 20 (3):199-204 doi:10.5020/18061230.2007.p199 https://doi.org/10.5020/1026

Saintrain, MVL. Vieira, APGF. Application of the Community Oral Health Indicator by Non-Dental Personnel and Its Contribution to Oral Healthcare. July 2012. PLoS ONE 7(7):e39733. DOI: 10.1371/journal.pone.0039733

Saintrain MVL, Saintrain SV, Sampaio EGM, Ferreira BSP, Nepomuceno TC, Frota MA, Vieira-Meyer APGF

Older adults’ dependence in activities of daily living: Implications for oral health. Public Health Nurs. 2018;00:1–9. https://doi.org/10.1111/phn.12529

Saintrain MVL, Bezerra TMM, Santos FS, Saintrain SV, Pequeno LL, Silva RM, Vieira-Meyer APGF. Subjective well-being and oral discomfort in older people. International Psychogeriatrics (2018), 30:10, 1509–1517 © International Psychogeriatric Association 2018 doi:10.1017/S1041610218000145

The following text was added to the manuscript as follows: Concerning the stability of the measures, regarding the Instrumental Activities of Daily Living (IADL) developed by Lawton and Brody and adapted to Brazilian context, almost perfect agreement has been shown, both for the reliability (Ricc=0.89) as for the objectivity (Ricc=0.80) (Santos & Virtuoso Júnior, 2008).

I think the discussion of recommendations with regards to oral health for patients who are dependent (p. 13) may be rather simplistic. Given that such patients are dependent on others (caregivers), it seems like one explanation would be that the dependence/disability is the cause of the poor oral hygiene and oral health, and that the caregivers, might need intervention as much as the patient… That reasoning does come out on page 14.

We fully agree with the reviewer. We have added some extra points in the text, as follows: “Furthermore, due to the common dependence level of these patients, including for selfcare, it is important to motivate and train caregivers on oral health care. Oral health issues may commonly be left, by caregivers, to a second plane of importance due the challenges brought by other general health issues.” 

I was confused about the discussion at the bottom of page 15, where you talk about a program tailored for caregivers – are you saying that the positive findings for that study support an underlying premise that poor oral health in dependent patients may be due to inadequate oral hygiene provided by caregivers? That is a logical conclusion, given that the dependent patient is dependent for care on others – just need to explain what you mean more clearly…

We were actually trying to state that a tailored educational program focusing on caregivers have a positive impact on dependent elder’s oral health. We have adjusted the manuscript in order to clarify this issue. “The findings of the present research showed that functional dependence in IADL is associated with oral hygiene and behaviors that maintain oral health, and therefore, is an important health indicator in older adults. Oral health problems can have profound effects on general health and well-being. Furthermore, oral pain and problems with eating, chewing, smiling and communicating due to missing, discolored or damaged teeth have a major impact on functional ability and older people’s daily lives [14]. Thus, as mentioned before, investments in oral health care through caregiver’s motivation and training may have a positive impact of elder’s health. In this regard, researchers have demonstrated that an educational program tailored to caregivers had a positive impact on the oral health of dependent older adults, observed by the improved oral hygiene parameters, such as plaque index and tongue coating [33].”

Reviewer #2: Thank you for your submission and yes I agree that we need to assess the status of older population in order to better serve their needs.

You mentioned that some of your questionnaires are validated what about the others, where they validated? If not, how can you justify their use and reliability?

We only utilized questionnaires that were either validated or broadly used. Some of the questionnaires, despite not being validated in Brazil, have been broadly used in published research performed in the country. 

-Were your examiners calibrated? If yes, how and how is their agreement. If not, how reliable is your data?

The examiners were properly trained by the co-author MVLS to apply the questionnaires and the ICSB, following adequate protocols for this purpose. The ICSB was created and validated by two co-authors of this manuscript (MVLS and APGFVM), which assures that the training was performed as advocated by the creators of the indicator and that the data is reliable. Furthermore, the examiners (n=?) were closely accompanied by the co-author MVLS during data collection. The following text was inserted in the manuscript “The examinations were performed under natural light with the help of a wooden spatula (tongue depressor) – examiners wore personal protective equipment. The two examiners who performed this evaluation were trained and closely accompanied by the co-author and gold standard (one of the ICSB creators) MVLS. The minimal accepted kappa values for intra and inter examiner was 0.8.” 

-Where you able to match data from supposes? How does marriage play a role in assessing dependance?

Regarding the point raised by the reviewer, we have added some information in the discussion part of the manuscript: “A relationship was observed between marital status and AIVD independence. It is not clear the path by which marital status plays a role on dependence, however, a possibility is related to the elders’ quality of life and mental status. According to Gutiérrez-Vega et al. (2018), married older adults had the highest quality of life in social relationships when compared to other groups. They state that marital status may play an important role when analyzing quality of life among older adults, suggesting that being married may offer a protective mechanism against depressive symptoms and therefore against mental illnesses during late adulthood.”

Gutiérrez-Vega M, Esparza-Del Villar OA, Carrillo-Saucedo IC, Montañez-Alvarado P. The Possible Protective Effect of Marital Status in Quality of Life Among Elders in a U.S.-Mexico Border City. Community Ment Health J. 2018;54(4):480-484. doi:10.1007/s10597-017-0166-z

-There were some instances that you used causation as a term instead of association. Causation could not be established at this stage so please correct accordingly.

We thank the reviewer for bringing up this important point. We have adjusted the manuscript in order to clarified this issue. The objective of the present study was to evaluate the existence of an association between oral health and AIVD, which we found. At this point, we cannot imply causality in this association, and the text was adjusted in this manner. 

-While you addressed generalizability as a limitation but it was not part of your conclusion statement. Please modify your conclusion so that it is clear that it does not apply to all population.

Thank you for calling our attention for this issue. We have amend the conclusion in order to clarify that the findings relate to elders living in the city of Fortaleza.

“Our findings demonstrated that functional dependence in instrumental activities of daily living can be considered an indicator of oral health status in older adults living in Fortaleza. Functional disability was associated with older age, cardiovascular diseases, speech problems, chewing and swallowing difficulties, use of medication and brushing deficiency. Therefore, health promotion and disease prevention and rehabilitation actions should be carried out at all levels of health care considering the needs of this population group in order to improve oral hygiene and oral health-friendly behaviors”.

6. PLOS authors have the option to publish the peer review history of their article (what does this mean?). If published, this will include your full peer review and any attached files.

Do you want your identity to be public for this peer review? For information about this choice, including consent withdrawal, please see our Privacy Policy.

Reviewer #1: No

Reviewer #2: No

---

## [Decision Letter · Decision Letter 1]

24 Nov 2020

PONE-D-20-18747R1

Dependence in instrumental activities of daily living and its implications for older adults’ oral health

PLOS ONE

Dear Dr. Saintrain,

Thank you for submitting your manuscript to PLOS ONE. After careful consideration, we feel that it has merit but does not fully meet PLOS ONE’s publication criteria as it currently stands. Therefore, we invite you to submit a revised version of the manuscript that addresses the points raised during the review process.

We look forward to receiving your revised manuscript.

Kind regards,

Frédéric Denis, Ph.D.

Academic Editor

PLOS ONE

Reviewers' comments:

Reviewer's Responses to Questions

**Comments to the Author**

1. If the authors have adequately addressed your comments raised in a previous round of review and you feel that this manuscript is now acceptable for publication, you may indicate that here to bypass the “Comments to the Author” section, enter your conflict of interest statement in the “Confidential to Editor” section, and submit your "Accept" recommendation.

Reviewer #2: All comments have been addressed

Reviewer #3: All comments have been addressed

2. Is the manuscript technically sound, and do the data support the conclusions?

Reviewer #2: Yes

Reviewer #3: Partly

3. Has the statistical analysis been performed appropriately and rigorously? 

Reviewer #2: I Don't Know

Reviewer #3: Yes

4. Have the authors made all data underlying the findings in their manuscript fully available?

Reviewer #2: Yes

Reviewer #3: Yes

5. Is the manuscript presented in an intelligible fashion and written in standard English?

Reviewer #2: Yes

Reviewer #3: No

6. Review Comments to the Author

Reviewer #2: Dear Author(s),

Thank you for addressing my comments. it is my decision to accept your paper after review of your response.

Reviewer #3: Background section

The background summary of research leading to the study’s aim is not entirely clear as written. While the study’s aim is to explore associations between degrees of functionality among aging adults in relationship with indicators of oral health, there is a lot of additional clinical considerations that are presented but in a disjointed manner. For example, discussion of chronic disease, edentulism, periodontal disease, disability, and disability are mentioned, but the connection the team of authors is making is not presented in a logical manner. The statement: understanding dimensions of the negative impact of functional impairment caused by poor oral and general health is critical for providing adequate healthcare and health education. Yet, given the cross sectional study design, determination of causality is not possible. A clear presentation of how oral health has risen to be a global health priority (is this the first WHO global health NCD agenda to include oral health?) does not directly convey if the status of addressing oral health needs among older adults is only now gaining recognition and the impetus behind it. Are there epidemiologic studies to quantify years of life lost among individuals with periodontal disease, edentulism, medication-related xerostomia? This support would strengthen the main tenet of this research and as presented, the value of answering the research question is not entirely clear and convincing.

Within the Methods section, the first sentence is very convoluted:

All procedures performed in studies involving human participants were in accordance with the Institutional Review Board (Approval No. 200/2009) and the national research committee and with the 1964 Helsinki declaration and its later amendments or comparable ethical standards.

It includes 4 uses of ‘and’.

The second paragraph in this section (p5) is a single sentence. Typically, a developed paragraph will have 3 sentences.

Again, in the sentence:

All the participants were informed of the objectives and procedures of the study and anonymity of participants and confidentiality of information were guaranteed.

The quality of writing could be improved with additional proofreading.

There is no need to include the STROBE checklist as a supplemental file nor to reference it within the abstract. It is a recognized checklist that can be mentioned within the manuscript and readers can seek details if interested.

Concerning the instrumentation-

The statement:

The semi-structured questionnaire addressing sociodemographic data (age, sex, race, marital status, education and income), general health (poor, fair, very good), self-reported 6 diseases and oral discomfort data (dry mouth, difficulty in chewing and swallowing food, taste of food, burning mouth sensation, pain without apparent reason, swelling of the mouth).

1. Was a validated general health scale not considered (vs semi-structured questionnaire)?

2. Was regimen of medication not assessed? As types of medications is related to xerostomia, for example.

3. What about water source and fluoride exposure?

4. Why not ask about nutritional and behavioral risks, such as use of alcohol and/or tobacco products? Soft-drink consumption? Any regional/national drinks that may affect tooth enamel (such as carbonated drinks, kombucha, etc?)

For the next statement about the ICSB-

5. And items within the Community Oral Health Indicator (Indicador Comunitário de Saúde Bucal – ICSB) may be scientifically validated for specificity, sensitivity, but overall, within the context of overall health and functioning-how do the items correspond in terms of functioning/potential functional impairment? This validated indicator checks the number of teeth, visible dental cavities, presence of calculus, gingival inflammation, residual root, soft tissues injuries, use and need of dental prosthesis, use of toothbrush and toothpaste, and visits to the dentist [16]. The examinations were performed under natural light with the help of a wooden spatula (tongue depressor) – examiners wore personal protective equipment.

**what about flossing behavior? Use of water picks/syringes for flushing gums? Gingival bleeding? Fluoride exposure?

RESULTS section

1.The first statement:

Participants were 280 older people aged 60-96 years, with a mean age of 69.9 (SD±7.2) years.

Again, it is advisable to seek service from an external native English speaker, as the writing could be more fluid. For example, the study sample consisted of 280 older adults ranging from 60-96 years of age….

2. There is no frame of reference to indicate that your sample is reflective of the region overall-or if your study sample characteristics were skewed in any way.

3. The unit of ‘one minimum wage’ does not make sense and is included in presentation of your results (and within Table 1). It is not clear what this means.

4. Table 1 is missing closing parentheses within income values

5. There was no previous indication that alcohol and smoking behavior were assessed, yet results concerning these exposures are presented in this section.

6. There is no contextual information provided in previous sections of manuscript to indicate why voice changes would be a significant indicator of oral health status and how it is assessed. Not sure why gingival bleeding would not be part of the assessment.

7. I am assuming active caries (lesions)

8. For the indicator-pain with no apparent reason-is this general reported pain within the oral cavity-including palette, tongue? Does this include reported throat pain? Pain within bones (mandible? Sinuses?)

9. The statement:

In addition, older adults with cardiovascular disorders were 1.77 more likely to be dependent in IADL than those without cardiovascular disorders and older adults who did not use toothbrush were 2.57 times more likely to be dependent in IADL than those who used toothbrush.

Scientific research supports that periodontal disease is significantly associated with cardiovascular health, yet, presentation of this known relationship is not clear in the background section of paper. Also, there is a comma instead of a period for the value 1.77 included in Table 4.

Discussion section

1. The statement:

The present study stands out for showing that dependence in IADL causes harms to the oral health of older adults.

This statement cannot be made within the context of the study design. There is no ability to determine if IADL ‘causes’ any oral health manifestations as this study’s goal is to explore if significant associations between key variables exist.

2. Continuing in the first paragraph:

As in most epidemiological studies, there was a greater participation of women and a higher rate of women with dependence in IADL, but there was no significant difference between sexes.

Any citation?

3. Statement:

This finding differs from a study that demonstrated that women have a greater incidence of disability in IADL than men, and that this difference is maintained even after controlling for social vulnerability among women and presence of chronic diseases [2].

It is very important to provide greater context into your study sample-and how it reflects characteristics of the general population and if there was an oversample of women?

4. In the next paragraph, starting with “In the present study….”, results of various studies are cited-but it is important to note if these are aging studies conducted in Brazil specifically. The structures/clinical supports of older adult populations vary greater in terms of national context and this is not presented in terms of your discussion points.

5. Be consistent with use of acronyms-IADL should be used from the first time it is introduced. In the discussion section alone-the ADL/IADL is frequently spelled out

6. The background information regarding items included in your analyses would be very helpful if presented in the earliest section of your paper. For example:

Poor oral health in older people is mainly manifested in high rates of tooth loss, dental caries, periodontal diseases, xerostomia (dry mouth) and other conditions resulting from poor oral hygiene [30].

Conclusion section

1. The first sentence overstates value of study:

Our findings demonstrated that functional dependence in instrumental activities of daily living can be considered an indicator of oral health status in older adults.

While your study may yield results that associations exist; however, a more sophisticated subsequent study would be warranted to build upon and advance the cross-sectional study design utilized in this paper.

Given that this draft manuscript was a revise and resubmit, of which I have not previously reviewed, I do not think it is publication worthy in its current state. The language/writing does not clearly present the study parameters in a clear manner, and the value of the findings are not always accurate.

7. PLOS authors have the option to publish the peer review history of their article (what does this mean?). If published, this will include your full peer review and any attached files.

Reviewer #2: No

Reviewer #3: No

---

## [Author Response · Author response to Decision Letter 1]

4 Feb 2021

Response to Reviewers

Dear Editor,

We are happy to know that our manuscript is being considered for publication at PLOS ONE. We read the reviewers’ comments and adjusted the manuscript. Below we bring a point-by-point response to the reviewers. Please do not hesitate in contacting us if you need any further information. 

Best regards,

Maria Vieira Saintrain and Co-authors

Point-by-point response to reviewers’ comments

Reviewers' comments:

Reviewer's Responses to Questions

Comments to the Author

1. If the authors have adequately addressed your comments raised in a previous round of review and you feel that this manuscript is now acceptable for publication, you may indicate that here to bypass the “Comments to the Author” section, enter your conflict of interest statement in the “Confidential to Editor” section, and submit your "Accept" recommendation.

Reviewer #2: All comments have been addressed

Reviewer #3: All comments have been addressed

2. Is the manuscript technically sound, and do the data support the conclusions?

Reviewer #2: Yes

Reviewer #3: Partly

3. Has the statistical analysis been performed appropriately and rigorously?

Reviewer #2: I Don't Know

Reviewer #3: Yes

4. Have the authors made all data underlying the findings in their manuscript fully available?

Reviewer #2: Yes

Reviewer #3: Yes

5. Is the manuscript presented in an intelligible fashion and written in standard English?

Reviewer #2: Yes

Reviewer #3: No

Reviewer #3: Background section

The background summary of research leading to the study’s aim is not entirely clear as written. While the study’s aim is to explore associations between degrees of functionality among aging adults in relationship with indicators of oral health, there is a lot of additional clinical considerations that are presented but in a disjointed manner. For example, discussion of chronic disease, edentulism, periodontal disease, disability, and disability are mentioned, but the connection the team of authors is making is not presented in a logical manner. 

Thank you for your comments. We have reorganized the background section in order to clarify the points you raised. Changes are highlighted in the manuscript and transcribed below.

“The decline in functional capacity during aging leads to difficulties in performing basic activities of daily living – ADL (bathing, dressing, feeding, transferring/walking and sphincter control) and instrumental activities of daily living – IADL (keeping track of finances, using the telephone or computer, managing medication, shopping) [1], where IADL involve more complex levels of physical functioning and neuropsychological organization than ADL [2].

Disability has a huge impact on oral health care and the treatment of functionally dependent older adults is challenging due to their inability to perform oral hygiene and their need to use multiple drugs, which create an oral environment conducive to a rapid destruction of teeth [3, 4]. On the other hand, oral diseases represent a silent epidemic that disproportionately affects older people, with caries and periodontal diseases standing out as the two greatest health threats and the most common chronic diseases for the cause of tooth loss among adults [5], which in turn, has the potential to affect elders dependency, specially related with proper nutrition. For the WHO, a minimum of 20 teeth are necessary for proper oral function/nutrition [6]. 

Poor oral health in older people is mainly manifested in high rates of tooth loss, dental caries, periodontal diseases, xerostomia (dry mouth) and other conditions resulting from poor oral hygiene [7]. Oral diseases are undoubtedly a global public health problem linked to social and economic changes, and their ongoing neglect in global health policy highlights the need to address oral diseases among other non-communicable diseases (NCDs) as a global health priority [8].

According to the Global Burden of Disease Study 2016, oral diseases affected circa 3.58 billion people worldwide, and caries of the permanent teeth were the most prevalent of the conditions assessed. The cumulative burden of oral conditions dramatically increased between 1990 and 2015. The number of people with untreated oral conditions rose from 2.5 billion in 1990 to 3.5 billion in 2015, with a 64% increase in DALYs (disability-adjusted life year) due to oral conditions throughout the world [9]. Severe periodontal disease, which results mainly from poor oral hygiene and which can lead to tooth loss, was the 11th most prevalent disease globally [10]. Researchers have found that, in adjusted Poisson regression analysis, older age and functional disability were associated with edentulism [11]. Thus, it seems that oral health effects and is affected by elders’ functionality. 

 The increase in life expectancy has attracted growing interest in the effect of oral health on general health outcomes, especially those related to functionality, well-being and mortality [12]. Oral health and general health share common risk factors related to diet, the use of tobacco, and the excessive consumption of alcohol and the solutions to control oral disease are to be found through shared approaches with integrated chronic disease prevention [13].

 In this regard, public health solutions for oral diseases are most efficient when they are integrated with those for other NCDs and other national public health programs. The importance of oral health goals was first emphasized in 1981 by WHO as part of the program Health for All by the year 2000 [14, 15] and most recently the WHO, jointly with the FDI World Dental Federation and the International Association for Dental Research (IADR), formulated goals for oral health by the year 2020 [16].

Furthermore, the WHO Global Oral Health Program is aligned with the Global NCD agenda and the Shanghai Declaration on promoting health in the 2030 Agenda for Sustainable Development [17]. According to Petersen [13], the WHO Oral Health program gives priority to research helping correct the so called 10/90 gap which relates to the fact that only 10%of funding for global health research is allocated to health problems that affect 90% of the world population. The WHO Oral Health Program focuses on stimulating oral health research in the developed and developing world to reduce risk factors and the burden of oral disease, and to improve oral health systems and the effectiveness of community oral health programs. Building and strengthening research capacity in public healthcare highly recommended by WHO for effective control of disease and the socioeconomic development of any given country.

In Brazil, the rate of dependence in at least one ADL or IADL among older adults is 32.6% and the prevalence rate of exclusive dependence in IADL is 14.6% [18]. Population aging can be categorized into three broad functional groups: functionally independent, frail, and functionally dependent. Chronic and oral health diseases have a greater potential to impact frail and functionally dependent older people. However, it is important to understand that, although chronic illness is common among older adults, it does not necessarily lead to disability, but constitutes an important risk factor for its development [19]. 

In view of what was outlined above, there is an evident need for collaboration between the medical and dental fields and between professionals in both fields in order to make healthcare provision systems more effective [20]. Therefore, understanding the dimensions of the impact in terms of functional impairment caused by poor oral and general health is critical for the provision of adequate health care, communication and health education [21]. Given that, the objective of the present study was to assess the relationship between IADL and Oral Health of older people.”

The statement: understanding dimensions of the negative impact of functional impairment caused by poor oral and general health is critical for providing adequate healthcare and health education. Yet, given the cross sectional study design, determination of causality is not possible. A clear presentation of how oral health has risen to be a global health priority (is this the first WHO global health NCD agenda to include oral health?) does not directly convey if the status of addressing oral health needs among older adults is only now gaining recognition and the impetus behind it. 

More information was inserted in the introduction section in order to address this issue. This can be seen below and is highlighted in the manuscript.

“The increase in life expectancy has attracted growing interest in the effect of oral health on general health outcomes, especially those related to functionality, well-being and mortality [12]. Oral health and general health share common risk factors related to diet, the use of tobacco, and the excessive consumption of alcohol and the solutions to control oral disease are to be found through shared approaches with integrated chronic disease prevention [13].

 In this regard, public health solutions for oral diseases are most efficient when they are integrated with those for other NCDs and other national public health programs. The importance of oral health goals was first emphasized in 1981 by WHO as part of the program Health for All by the year 2000 [14, 15] and most recently the WHO, jointly with the FDI World Dental Federation and the International Association for Dental Research (IADR), formulated goals for oral health by the year 2020 [16].

Furthermore, the WHO Global Oral Health Program is aligned with the Global NCD agenda and the Shanghai Declaration on promoting health in the 2030 Agenda for Sustainable Development [17]. According to Petersen [13], the WHO Oral Health program gives priority to research helping correct the so called 10/90 gap which relates to the fact that only 10%of funding for global health research is allocated to health problems that affect 90% of the world population. The WHO Oral Health Program focuses on stimulating oral health research in the developed and developing world to reduce risk factors and the burden of oral disease, and to improve oral health systems and the effectiveness of community oral health programs. Building and strengthening research capacity in public healthcare highly recommended by WHO for effective control of disease and the socioeconomic development of any given country.” 

Are there epidemiologic studies to quantify years of life lost among individuals with periodontal disease, edentulism, medication-related xerostomia? This support would strengthen the main tenet of this research and as presented, the value of answering the research question is not entirely clear and convincing.

Yes, there is some data on DALYs related to oral health conditions. Those were inserted in the text and are presented below:

“According to the Global Burden of Disease Study 2016, oral diseases affected circa 3.58 billion people worldwide, and caries of the permanent teeth were the most prevalent of the conditions assessed. The cumulative burden of oral conditions dramatically increased between 1990 and 2015. The number of people with untreated oral conditions rose from 2.5 billion in 1990 to 3.5 billion in 2015, with a 64% increase in DALYs (disability-adjusted life year) due to oral conditions throughout the world [9]. Severe periodontal disease, which results mainly from poor oral hygiene and which can lead to tooth loss, was the 11th most prevalent disease globally [10]. Researchers have found that, in adjusted Poisson regression analysis, older age and functional disability were associated with edentulism [11]. Thus, it seems that oral health effects and is affected by elders’ functionality.” 

Within the Methods section, the first sentence is very convoluted:

All procedures performed in studies involving human participants were in accordance with the Institutional Review Board (Approval No. 200/2009) and the national research committee and with the 1964 Helsinki declaration and its later amendments or comparable ethical standards.

It includes 4 uses of ‘and’.

Thank you for point this out. We adjusted the text as follows: “All procedures performed in studies involving human participants were in accordance with the national research committee, as per approval No. 200/2009 from the University of Fortaleza Institutional Review Board. This committee follows the 1964 Helsinki declaration and its later amendments or comparable ethical standards. Informed consent was obtained from all individual participants included in the study.”

The second paragraph in this section (p5) is a single sentence. Typically, a developed paragraph will have 3 sentences.

Again, in the sentence:

All the participants were informed of the objectives and procedures of the study and anonymity of participants and confidentiality of information were guaranteed.

The quality of writing could be improved with additional proofreading.

There is no need to include the STROBE checklist as a supplemental file nor to reference it within the abstract. It is a recognized checklist that can be mentioned within the manuscript and readers can seek details if interested.

Thank you for your attention in all the details. We have adjusted the text as follows: “All the participants were informed of the objectives and procedures of the study. Participants anonymity and confidentiality of information were guaranteed. Data were individually collected and analyzed using:”

Concerning the instrumentation-

The statement:

The semi-structured questionnaire addressing sociodemographic data (age, sex, race, marital status, education and income), general health (poor, fair, very good), self-reported 6 diseases and oral discomfort data (dry mouth, difficulty in chewing and swallowing food, taste of food, burning mouth sensation, pain without apparent reason, swelling of the mouth).

1. Was a validated general health scale not considered (vs semi-structured questionnaire)?

None of the general health scale known by the authors addressed all the information we were interested in. Thus, we opted for the semi-structured questionnaire. Nevertheless, we based the oral health questions on the WHO Oral Health Questionnaire for Adults.

2. Was regimen of medication not assessed? As types of medications is related to xerostomia, for example.

Unfortunately, not. We agreed with the reviewer that this could’ve added to a broader analysis and discussion of the results. Nevertheless, we believe that the amount of information gathered in our study is robust for our study objective.

3. What about water source and fluoride exposure?

This information was not collected. However, Fortaleza has a fluoridated public water system. Nevertheless, the control of this fluoridation in the municipality or the water ingestion habits (mineral or public water) of the studied population is unknown.

4. Why not ask about nutritional and behavioral risks, such as use of alcohol and/or tobacco products? Soft-drink consumption? Any regional/national drinks that may affect tooth enamel (such as carbonated drinks, kombucha, etc?)

We did ask about tobacco and alcohol consumption, which is presented in table 2. Nutritional consumption is an interesting, yet complex, to analyze and point direct causality to oral health disease. 

5. For the next statement about the ICSB-And items within the Community Oral Health Indicator (Indicador Comunitário de Saúde Bucal – ICSB) may be scientifically validated for specificity, sensitivity, but overall, within the context of overall health and functioning-how do the items correspond in terms of functioning/potential functional impairment? This validated indicator checks the number of teeth, visible dental cavities, presence of calculus, gingival inflammation, residual root, soft tissues injuries, use and need of dental prosthesis, use of toothbrush and toothpaste, and visits to the dentist [16]. The examinations were performed under natural light with the help of a wooden spatula (tongue depressor) – examiners wore personal protective equipment.

**what about flossing behavior? Use of water picks/syringes for flushing gums? Gingival bleeding? Fluoride exposure?

According to the World Health Organization-WHO (1992), a minimum of 20 teeth are necessary for proper oral function/nutrition. A large number of missing teeth, dental decay, residual dental roots, tartar and gingival inflammation, among other oral issues, can affect the elders’ oral health and their ability to keep proper oral hygiene and their daily activities. Thus, the ICSB is able to identify issues with potential impact on elders’ AVD and AIVD.

Fluoride exposition through fluoridated dental paste was also investigated. However, flossing habits and the existence of gingival bleeding was not questioned. 

RESULTS section

1.The first statement:

Participants were 280 older people aged 60-96 years, with a mean age of 69.9 (SD±7.2) years.

Again, it is advisable to seek service from an external native English speaker, as the writing could be more fluid. For example, the study sample consisted of 280 older adults ranging from 60-96 years of age….

Thank you for your suggestion. English was revised and this statement now reads “The study sample consisted of 280 older adults ranging from 60-96 years of age, with a mean age of 69.9 (SD±7.2) years.”

2. There is no frame of reference to indicate that your sample is reflective of the region overall-or if your study sample characteristics were skewed in any way.

The information below was added to the text in order to clarify the sample calculation.

The sample size was estimated considering the number of older adults (N=242,430) in the city of Fortaleza in the year 2012 as reported in Brazil’s National Health System Database [18]. A minimum sample size was estimated considering an expected maximum proportion of 20%, a significance level of 5% (95% confidence interval), and a maximum permissible error of 5%.

3. The unit of ‘one minimum wage’ does not make sense and is included in presentation of your results (and within Table 1). It is not clear what this means.

Brazilian minimum wage is R$ 1,100.00 - approximately U$ 200.00. This information was added to table 1.

4. Table 1 is missing closing parentheses within income values

Thank you for pointing this up. We have adjusted the text. 

5. There was no previous indication that alcohol and smoking behavior were assessed, yet results concerning these exposures are presented in this section.

We have added information on this issue on the Materials and methods section as follows: “1. Semi-structured questionnaire addressing sociodemographic data (age, sex, race, marital status, education and income), general health (poor, fair, very good), self-reported diseases and oral discomfort data (dry mouth, difficulty in chewing and swallowing food, taste of food, burning mouth sensation, pain without apparent reason, swelling of the mouth), as well as alcohol and smoking habits.”.

6. There is no contextual information provided in previous sections of manuscript to indicate why voice changes would be a significant indicator of oral health status and how it is assessed. Not sure why gingival bleeding would not be part of the assessment.

The voice box (larynx) is made of cartilage, muscle and mucous membranes located at the top of the trachea and the base of the tongue. If the vocal cords become inflamed, develop growths or become paralyzed, they can't work properly, and one may develop a voice disorder. A voice change can be a sign of throat dehydration, Neurological disorders, or consequence of habits like alcohol consumption and tobacco usage. Furthermore, voice change can also be linked with oral discomfort. Gingival inflammation was assessed. We agree with the reviewer that gingival bleeding assessment could improve our study – we will definitely consider its inclusion in future studies. 

Changes were made in the manuscript, as follows:

“Furthermore, the increased PR estimates between dependence in IADL and difficulty chewing and swallowing (2.19), voice changes (2.22), edentulous (2.30), use of toothbrush (4.90), use of toothpaste (5.07) and visit to the dentist in the previous year (2.42) demonstrates a close relationship between disability and oral health preventive measures. Our voice is an extremely sensitive indicator of our general health and emotional status [31]. The voice box (larynx) is made of cartilage, muscle and mucous membranes located at the top of the trachea and the base of the tongue. If the vocal cords become inflamed, develop growths or become paralyzed, they can't work properly, and one may develop a voice disorder. A voice change can be a sign of throat dehydration, Neurological disorders, or consequence of habits like alcohol consumption and tobacco usage. Furthermore, voice change can also be linked with oral discomfort. Considering that dental caries and periodontal disease are associated with lifestyle, patients should be encouraged to visit the dentist regularly in addition to being provided with dental health guidance in each visit, as health education has short-term effects [32].”

7. I am assuming active caries (lesions)

For the ICSB, dental caries are active caries lesions or non-restored caries cavities. 

8. For the indicator-pain with no apparent reason-is this general reported pain within the oral cavity-including palette, tongue? Does this include reported throat pain? 

Pain within bones (mandible? Sinuses?)

The question was “do you feel pain for no apparent reason”? Thus, it includes anything that the responder believes to be pain without apparent reason.

9. The statement:

In addition, older adults with cardiovascular disorders were 1.77 more likely to be dependent in IADL than those without cardiovascular disorders and older adults who did not use toothbrush were 2.57 times more likely to be dependent in IADL than those who used toothbrush. Scientific research supports that periodontal disease is significantly associated with cardiovascular health, yet, presentation of this known relationship is not clear in the background section of paper. Also, there is a comma instead of a period for the value 1.77 included in Table 4

Thank you for pointing that up. We have adjusted the text accordingly. 

Discussion section

1. The statement:

The present study stands out for showing that dependence in IADL causes harms to the oral health of older adults.

This statement cannot be made within the context of the study design. There is no ability to determine if IADL ‘causes’ any oral health manifestations as this study’s goal is to explore if significant associations between key variables exist.

We fully agree with the reviewer. The statement was adjusted as follows: “The present study stands out for showing a relationship between dependence in IADL and oral health of older adults”. 

2. Continuing in the first paragraph:

As in most epidemiological studies, there was a greater participation of women and a higher rate of women with dependence in IADL, but there was no significant difference between sexes.

Any citation?

We have also adjusted the statement in order to clarify what we wanted to say. The changes are highlighted in the manuscript. 

“Gender did not appear to have any influence on IADL dependency.”

3. Statement:

This finding differs from a study that demonstrated that women have a greater incidence of disability in IADL than men, and that this difference is maintained even after controlling for social vulnerability among women and presence of chronic diseases [2].

It is very important to provide greater context into your study sample-and how it reflects characteristics of the general population and if there was an oversample of women?

Changes were done in the manuscript in order to address this issue: “The present study stands out for showing a relationship between dependence in IADL and oral health of older adults. Gender did not appear to have any influence on IADL dependency. This finding differs from a study that demonstrated that women have a greater incidence of disability in IADL than men, and that this difference is maintained even after controlling for social vulnerability among women and presence of chronic diseases [2]. According to the latest IBGE census, women represent 60,7% of the elder population in Fortaleza [26]. Thus, through authors’’ perspective, our female sample (65%) it comparable with the city reality and did not seem to influence our findings regarding IADL in the elder population.”

4. In the next paragraph, starting with “In the present study….”, results of various studies are cited-but it is important to note if these are aging studies conducted in Brazil specifically. The structures/clinical supports of older adult populations vary greater in terms of national context and this is not presented in terms of your discussion points.

We have modified the manuscript in order to follow the reviewers’ suggestion: “In the present study, the most prevalent age group and the mean age are similar to those of another Brazilian study, where the mean age was 70.04 (±7.89) years and most people were aged 60-69 years [27]. The percentage of 13.2% of older people with different degrees of dependence in their IADL is similar to that reported in the literature for Brazil [28, 18]. However, a study shows that 46.3% of a population of 289 older people had moderate to severe dependence in IADL [29]. It is important to know that the characteristics identified regarding dependence in IADL suggest a complex causal relationship that points to the need for preventive actions specifically designed to address different factors and improve older people’s quality of life [27]. Nonetheless, it is important to point out that all the studies just mentioned were carried out in Brazil. As the structures, clinical supports and family care of older adult populations may vary in terms of national context and culture, it is important to have these issues in mind when extrapolating the study findings.”

5. Be consistent with use of acronyms-IADL should be used from the first time it is introduced. In the discussion section alone-the ADL/IADL is frequently spelled out

Thank you for pointing that out. We have revised the manuscript and adjusted it according to your suggestion. 

6. The background information regarding items included in your analyses would be very helpful if presented in the earliest section of your paper. For example:

Poor oral health in older people is mainly manifested in high rates of tooth loss, dental caries, periodontal diseases, xerostomia (dry mouth) and other conditions resulting from poor oral hygiene [30].

Thank you for your comment. We have inserted this information in the introduction section. 

Conclusion section

1. The first sentence overstates value of study:

Our findings demonstrated that functional dependence in instrumental activities of daily living can be considered an indicator of oral health status in older adults.

While your study may yield results that associations exist; however, a more sophisticated subsequent study would be warranted to build upon and advance the cross-sectional study design utilized in this paper.

We have adjusted the conclusion statement as suggested by the reviewer. 

“Our findings demonstrated that functional dependence in IADL is related to oral health status in older Brazilian adults. Functional disability was associated with older age, cardiovascular diseases, speech problems, chewing and swallowing difficulties, use of medication and brushing deficiency. Therefore, health promotion and disease prevention and rehabilitation actions should be carried out at all levels of health care considering the needs of this population group in order to improve oral hygiene and oral health-friendly behaviors.”

Given that this draft manuscript was a revise and resubmit, of which I have not previously reviewed, I do not think it is publication worthy in its current state. The language/writing does not clearly present the study parameters in a clear manner, and the value of the findings are not always accurate.

We hope that the changes/adjustments made in the manuscript make you reconsider the above statement. We are very pleased that our manuscript is under consideration at PLOS ONE and willing to make any further adjustments if necessary.

7. PLOS authors have the option to publish the peer review history of their article (what does this mean?). If published, this will include your full peer review and any attached files.

Do you want your identity to be public for this peer review? For information about this choice, including consent withdrawal, please see our Privacy Policy.

Reviewer #2: No

Reviewer #3: No

---

## [Editor Report · Decision Letter 2]

16 Mar 2021

Dependence in instrumental activities of daily living and its implications for older adults’ oral health

PONE-D-20-18747R2

Dear Dr. Saintrain,

We’re pleased to inform you that your manuscript has been judged scientifically suitable for publication and will be formally accepted for publication once it meets all outstanding technical requirements.

Kind regards,

Frédéric Denis, Ph.D.

Academic Editor

PLOS ONE
---

## [Editor Report · Acceptance letter]

22 Mar 2021

PONE-D-20-18747R2 

Dependence in instrumental activities of daily living and its implications for older adults’ oral health 

Dear Dr. Saintrain:

I'm pleased to inform you that your manuscript has been deemed suitable for publication in PLOS ONE. Congratulations! Your manuscript is now with our production department. 

Kind regards, 

on behalf of

Dr. Frédéric Denis 

Academic Editor

PLOS ONE